# Perceptions of carbon dioxide emission reductions and future warming among climate experts
Seth Wynes [1,2] ✉, Steven J. Davis [3], Mitchell Dickau[1], Susan Ly [1], Edward Maibach [4], Joeri Rogelj [5,6], Kirsten Zickfeld [7] & H. Damon Matthews [1]

The Intergovernmental Panel on Climate Change (IPCC) employs emission scenarios to explore a range of future climate outcomes but refrains from assigning probabilities to individual scenarios. However, IPCC authors have their own views on the likelihood of different climate outcomes, which are valuable to understand because authors possess both expert insight and considerable influence. Here we report the results of a survey of 211 IPCC authors about the likelihood of four key climate outcomes. We found that most authors are skeptical that warming will be limited to the Paris targets of well below 2 °C, but are more optimistic that net zero $CO_2$ emissions will be reached during the second half of this century. When asked about the beliefs of their peers, author responses showed strong correlations between personal and peer beliefs, suggesting that participants with extreme beliefs perceive their own estimates as closer to the community average than they actually are.

The IPCC uses emissions scenarios to assist policymakers in understanding potential future climate outcomes. These range from storylines where humanity limits warming through rapid and extensive societal climate action, to futures where planetary warming surpasses 4 °C in 2100 and continues to rise thereafter[1]. Even within a single emissions scenario, varying climate sensitivity could result in substantially different warming levels. The IPCC chooses to not describe the probability of different scenarios occurring, which is consistent with concerns that doing so would understate true levels of uncertainty[2]. However, providing information about the likelihood of different future outcomes would have real benefits to engineers, planners, and policymakers[3–5]. Furthermore, the absence of likelihood information about future climate outcomes means that the responsibility to select the most relevant scenarios for impacts and adaptation planning is transferred from climate scientists to policymakers[6,7].

While the IPCC community has opted against estimating likelihoods associated with future emission scenarios, it does indicate that the worst-case scenarios for future climate change (specifically the scenario leading to 8.5 W/m$^2$ radiative forcing in 2100, RCP8.5, SSP5-8.5) have become implausible[8]. At the same time, the chance of achieving best case scenarios is declining in lockstep with the dwindling carbon budget[9]. Climate scientists who model future climate impacts (especially those in Working Group 2 on

impacts and adaptation) must select scenarios as inputs, and may be influenced by their perceptions of relative likelihoods. However, it is unclear whether there is agreement across working groups about which scenarios are perceived to be more or less likely.

The assumptions and modeling choices made by the IPCC and the climate science community are highly influential, shaping political decisions and potentially influencing future technological pathways[10,11]. Furthermore, many IPCC authors engage both directly and indirectly (through media interviews) in efforts to share their knowledge with the public and policymakers[12], which sometimes includes sharing their professional beliefs on a range of speculative topics, including the feasibility of achieving temperature targets[13]. Since IPCC authorship is treated as a credential, with lead authors benefiting from improved reputation and the ability to influence policy[14], the beliefs of authors regarding key future climate outcomes have considerable value and are worth investigating and understanding.

Researchers may develop their personal beliefs of future climate outcomes in a variety of ways. They might interpret the evidence of data and models themselves, or could develop opinions reading literature authored by others (including the IPCC reports themselves). But they may also be influenced by their perceptions of peer beliefs, sometimes known as second-order beliefs[15,16]. This could potentially be problematic if climate scientists

[1]Department of Geography, Planning & Environment, Concordia University, Montréal, QC, Canada. [2]Department of Geography and Environmental Management, University of Waterloo, Waterloo, ON, Canada. [3]Dept. of Earth System Science, Stanford University, Stanford, CA, USA. [4]Department of Communication, George Mason University, Fairfax, VA, USA. [5]Grantham Institute Climate Change and Environment, Centre for Environmental Policy, Imperial College London, London, UK. [6]Energy Climate and Environment Programme, International Institute for Applied Systems Analysis, Laxenburg, Austria. [7]Department of Geography, Simon Fraser University, Burnaby, BC, Canada. ✉e-mail: swynes@uwaterloo.ca

feel pressured to abide by norms of restraint and therefore publicly downplay negative, dramatic findings[17]. If this then caused other members of the community to underestimate the likelihood of negative outcomes, they might in turn rely on those faulty assumptions in their own research decisions or public communications. Conversely, scientists may also take the opposite approach: presenting information to the public about the potential for extreme future climate impacts in a way that is intended to incentivize action (IPCC reports buffer against such influence through their elaborate expert and government review cycles).

Here we present the results of a survey of IPCC authors to assess their expectations of the progression of mitigation efforts and the resulting climate changes that will occur this century. Our survey provides insights into three key questions: Which climate outcomes do members of the IPCC community believe are most likely? Do these beliefs vary by field of expertise? And do authors correctly understand the beliefs of their peers? To inform these questions, we asked IPCC authors for their first and second-order beliefs on four future climate outcomes: maximum global warming by the year 2100, likelihood of global temperatures reaching 3 °C by 2100, anticipated year that net zero global $CO_2$ emissions will be achieved, and rate of carbon dioxide removal (CDR) in 2050. Asking for estimates of four future climate outcomes had two benefits: it increased certainty in the understanding of the overall outlook of participants, and also illuminated differences in the way that IPCC authors understand potential futures. For instance, if participants anticipate high temperatures by 2100, but also anticipate net zero $CO_2$ being reached relatively early in the century, it could indicate beliefs about the difficulty in mitigating non-$CO_2$ gases which might be explored further in a later survey. Indeed, looking beyond this study we are interested in how authors' beliefs change over time and built the survey with a goal of regular updates.

We recruited participants for our online survey on perceptions of future climate outcomes from lists of authors posted by the IPCC for the AR6 cycle, which ranges from 2018 to 2023. 211 authors completed the survey for a conservative response rate of 23% (a somewhat lower response rate than two other surveys of IPCC contributors[14,18]). Respondents included representatives from every continent and every report issued by the IPCC since the Special Report on Global Warming of 1.5 °C (see Supplementary Table 1 for more on the representativeness of the sample).

### Pre-registered hypotheses

Before distributing the survey, we preregistered two hypotheses (see Data Availability Statement):

H1) There will be a positive correlation between estimates of peer beliefs and personal estimates of future climate outcomes.

This hypothesis was based on research in other domains demonstrating a "false consensus effect", where people believe their own judgments to be common and opposing judgments to be uncommon[19,20].

H2) Researchers with a focus on climate solutions (Working Group 3) will have more optimistic perceptions of future climate outcomes than those who work on climate impacts and adaptation (Working Group 2).

We expected that researchers working on climate solutions would have greater familiarity with recent literature suggesting that the worst-case climate outcomes have become increasingly unlikely[21,22] and greater familiarity with the rapidly improving solution set. Furthermore, because participants were generating rapid responses rather than carefully crafted analysis, we would expect them to rely on the availability heuristic (where people judge the probability of events by how readily they come to mind[23]) which would make researchers who work on climate solutions more prone to accessing optimistic estimates.

## Results

### Authors are skeptical of reaching global temperature targets

Few participants believed, given currently available evidence, that the global community will succeed in limiting global warming to the well-below 2 °C temperature goal of the Paris Agreement (Fig. 1). 86% of participants estimated maximum global warming of greater than 2 °C by or before the year 2100 (med = 2.7 °C) while 58% of the sample believed that there was at least a 50% chance of reaching or exceeding 3 °C by or before 2100 (med = 50%). These participant estimates are consistent with modeling outputs of the climate response to current national climate policies (2.7 °C by the year 2100[24]), and are generally higher than estimates of the anticipated temperature outcome associated with countries meeting their stated emissions targets.

However, when asked about the year that net zero $CO_2$ emissions might be achieved, participants were more optimistic: 66% of participants believed that net zero would be achieved before 2085 (med = 2075), which is consistent with future emission scenarios that limit warming to less than 2 °C. Participants were also optimistic about the potential for $CO_2$ removal: the median response was 5 $GtCO_2$/yr of CDR at 2050, which is believed to be at the lower end of the annual CDR rate that is likely necessary to achieve the Paris targets (estimated at between ~5 and 10 $GtCO_2$/yr at minimum[25]).

Although we have presented unweighted measures here, we also checked to see if issues with the representativeness of our survey affected the results. We

**Fig. 1 | Density plot showing subjective estimates of future climate outcomes in four categories.**
**a** Predicted maximum warming by or before 2100. Vertical line indicates median current trajectory according to existing policies (2.7 °C) and national pledges and targets (2.1 °C) while range indicates upper and lower bounds[24]. One data point with maximum warming estimated at 10 °C is outside of the range plotted here. **b** Likelihood that Earth will experience 3 °C of warming or more by or before 2100. **c** Year that human $CO_2$ emissions will reach net zero globally with dashed line at 2050 indicating a net zero year consistent with limiting global warming to 1.5 °C and solid line (at 2082.5) consistent with limiting warming to 2 °C[1]. **d** Estimated rate of carbon dioxide removal in 2050 with gray region indicating range consistent with achieving Paris Targets[25].

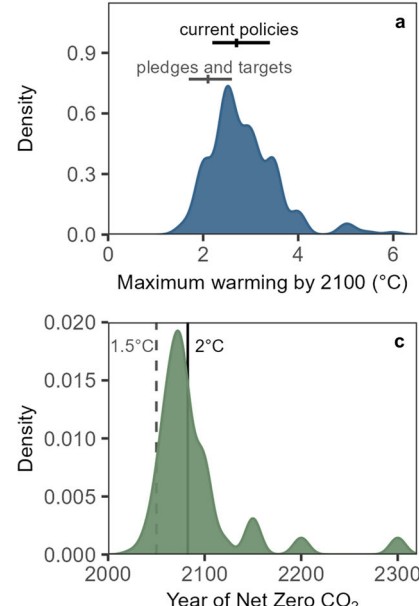
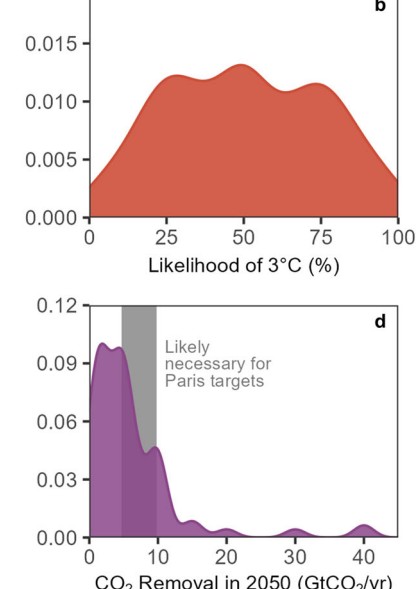

**Fig. 2 | Scatterplot of self versus peer perceptions of four future climate outcomes. a** Estimates of maximum warming by 2100 (°C). **b** Estimates of likelihood of 3 °C as a percentage. **c** Estimates of year of net zero $CO_2$. **d** Estimates of $CO_2$ removal in 2050 ($GtCO_2$/year). One value in (**a**) with maximum warming of 10 °C not visible.

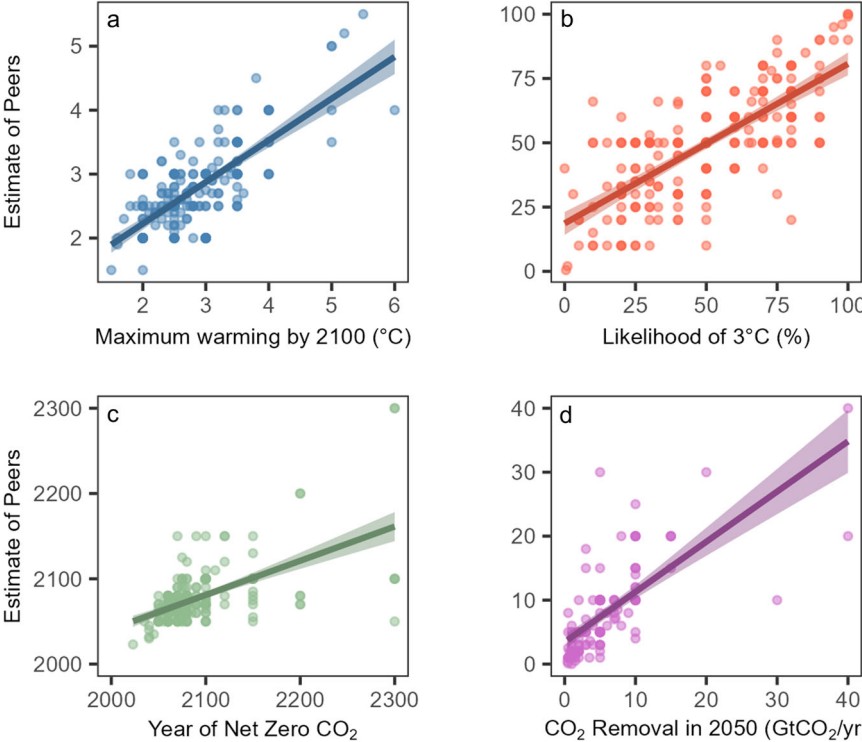

found that when our sample is weighted to reflect the demographics of the IPCC population as a whole (using proportional iterative weighting, or raking, according to gender, the continent of citizenship, and IPCC Working Group), the mean responses do not fluctuate substantially. For instance, the unweighted mean of estimated year of net zero $CO_2$ is 2090, whereas the weighted mean is 2088 (see Supplementary Table 2 for full results).

## Responses raise questions about the temperature outcome of net zero emissions

A majority of participants estimated maximum warming by 2100 to be greater than 2 °C, but most also estimated that net zero $CO_2$ would be achieved before 2100. This means that participant estimates of year of net zero $CO_2$ are reasonably aligned with IPCC scenarios that are expected to limit warming to less than 2 °C, but the same participants generally expect warming to exceed 2 °C. Consequently, when we plot net zero $CO_2$ scenarios reviewed by the IPCC versus end of century warming, there is limited overlap with author responses to these same questions (Supplementary Fig. 1). Furthermore, this pattern is not isolated to a single working group (Supplementary Fig. 2). This difference between scenario projections and author expectations of net zero years vs. temperature outcomes raises interesting questions about IPCC author understanding of or confidence in IPCC scenarios and their expected climate response, whether the emissions pathways included in the IPCC scenario assessment cover the pathways that authors other than scenario modelers imagine, or the clarity, interpretation and understanding of the survey questions.

## Personal and peer beliefs are closely related

Personal beliefs about future climate outcomes and perceptions of peer beliefs (second-order beliefs) were closely correlated, providing robust evidence for H1 (Fig. 2). The correlation between self and peer predictions was strong for predicted maximum warming by 2100 ($r_s = 0.62$, $p < 0.001$), strong for likelihood of 3 °C ($r_s = 0.72$, $p < 0.001$), moderate for year of net zero $CO_2$ ($r_s = 0.55$, $p < 0.001$), and strong for rate of CDR in 2050 ($r_s = 0.78$, $p < 0.001$).

We registered no hypothesis regarding differences between personal (first-order) and peer (second-order) estimates, but in exploratory tests we found no significant difference between the personal estimate (med = 2.7 °C)

and estimate of peer beliefs (med = 2.7 °C) for maximum global warming by 2100 ($p = 0.06$) or between personal (med = 50%) and peer estimate (med = 50%) for likelihood of reaching 3 °C ($p = 0.73$). The difference between personal (med = 2075) and peer estimates (med = 2070) for year of net zero $CO_2$ was significant ($p < 0.001$) as was the difference between personal (med = 5) and peer estimates (med = 6) for rate of CDR in 2050 ($p < 0.001$)(Supplementary Fig 3). This suggests that participants slightly overestimated the optimism of their peers with respect to their estimates of net zero year and the amount of CDR at 2050.

## Differences between expert groups are limited

We hypothesized that respondents with a focus on solutions, whose expertise is related to Working Group 3 (WG3) would have more optimistic responses than those working on adaptation and impacts (Working Group 2—WG2). We found limited evidence to support this hypothesis (H2). A Kruskal-Wallis test did find significant differences for the estimates of likelihood of reaching 3 °C ($H = 10.4$, $p = 0.006$) between the groups (WG1 med = 50, WG2 med = 60, WG3 med = 40) with a Bonferonni-adjusted pairwise Wilcox test specifically showing differences between WG2 and WG3 ($p = 0.02$). Differences between working groups for predicted maximum warming by 2100 ($H = 3.03$, $p = 0.22$), year of net zero $CO_2$ ($H = 1.46$, $p = 0.482$), and rate of CDR in 2050 ($H = 0.31$, $p = 0.857$) were not significant (Fig. 3). As a robustness check we performed the same tests used to evaluate H1 and H2 on the data with outliers still included and found no substantial changes (e.g., no result that was statistically significant became insignificant etc.) (Supplementary Results).

We had no hypothesis regarding whether some working groups would have a higher correlation between their first and secondary beliefs than others. However, we tested working group-specific correlations and found that all working groups displayed similar correlations between primary and secondary beliefs on all four future climate outcomes (Supplementary Table 3). The largest differences were observed between Working Group 2 and Working Group 3, but in a statistical comparison of those correlations, none were found to be significantly different. We also did not register a hypothesis regarding differences between continents, but have made a breakdown of results by continent available for the interested reader (Supplementary Table 4).

**Fig. 3 | Violin plot of participant beliefs of four future climate outcomes grouped according to self-described expertise. a** Estimates of maximum warming by 2100 (°C). **b** Estimates of likelihood of 3 °C as a percentage. **c** Estimates of year of net zero $CO_2$. **d** Estimates of $CO_2$ removal in 2050 (GtCO2/year). Working Group 1 (WG1) studies the Physical Science Basis, Working Group 2 (WG2) studies impacts, adaptation and vulnerability, while Working Group 3 (WG3) studies the mitigation of climate change.

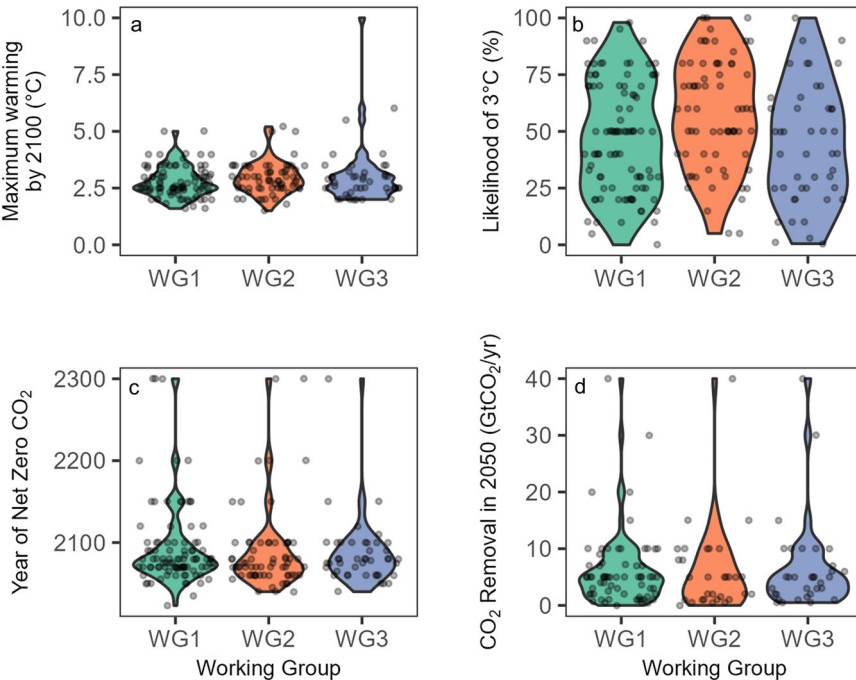

## Discussion

IPCC authors play a critical role communicating climate science information to policy makers and the public, so it is important to understand the personal beliefs that inform their thinking and how those beliefs are formed. In this study we found that IPCC authors are skeptical, given current evidence, that warming will be limited to 1.5 °C or well below 2 °C rise in temperature compared to pre-industrial averages. This is an interesting finding in light of independent estimates that suggest a median end-of-century temperature rise of 2.7 °C under existing national climate policies, or 2.1 °C based on a scenario in which all nations achieve their current pledges and targets[24]. The large majority (86%) of our respondents believed that warming would exceed 2 °C this century, and 58% of our sample believed there was at least a 50% likelihood of exceeding 3 °C. This suggests that IPCC author responses are closely aligned with a projection of current emission trends in the absence of widespread increases in national climate ambition. This reflects a more conservative attitude towards how climate measures will continue beyond their current target years (often set for 2025 or 2030), a skepticism that nations will achieve their stated climate targets, or a belief that the climate response to future emissions will be more extreme than what is estimated by current models.

A previous survey found 60% of Working Group 1 authors believed that warming of 3 °C or more is likely by 2100[18], while another found that 77% of IPCC authors and editors expected at least 2.5 °C in the same timeframe[26]. Our results corroborate that there is a widespread belief among IPCC authors that substantial warming is likely before 2100, but also extends these findings by asking about additional climate outcomes, and shedding light on second order beliefs. Other surveys of IPCC authors have also been conducted, focusing for instance on gender bias in the IPCC[14] and science communication[12].

### Support for preregistered hypotheses

In line with our expectations (H1), we found strong correlations between IPCC authors' beliefs of future climate outcomes, and their beliefs about what their peers believe. This is consistent with other research which shows that people tend to overestimate the degree to which others share their beliefs—sometimes referred to as a false-consensus effect[19,27]. The correlations we observed (ranging from $r_s = 0.55$ to $r_s = 0.78$) were towards the higher end of correlations reported for this phenomenon in a meta-analysis

of 115 studies, and well above the average of $r = 0.31$[20]. One consequence of this strong relationship is that participants who hold beliefs at either end of the distribution (e.g., those who believe that warming will be either very high or very limited) tend to incorrectly believe that their own perceptions are close to the community average. This could have consequences for the confidence with which communicators speak on behalf of the community.

One potential critique of this finding is that while we asked participants to estimate the average response of their peers, participants may have had in mind the actual population of IPCC researchers, and yet our sample is not perfectly representative of the population. We tried to address this by weighting the sample and found that weighted means of future climate estimates were remarkably similar to unweighted means (Supplementary Table 2). Realistically however, we would acknowledge that the more likely cognitive process for a participant trying to estimate the beliefs of the entire population would be to rely on the beliefs of near-peers, whose ideas would be more cognitively available than objective indicators (which are rare outside of this study).

We also hypothesized (H2) that researchers focusing on solutions (WG3) would be more optimistic than those focusing on impacts and adaptation (WG2). We only found evidence in favor of this hypothesis in responses to one of our four questions (likelihood of reaching 3 °C). First, this indicates that while IPCC authors may disagree on the likelihood of future scenarios with high levels of warming (such as RCP 8.5), these beliefs do not seem to differ significantly by working group. Second, since differences between expert groups were small and inconsistent, we do not have evidence that biased second-order perceptions are due to the isolation of researchers within groups of related expertise. Instead, a more likely explanation is that the strong correlation between first-order and second-order beliefs is due to participants engaging in a common heuristic: projecting their own understanding onto the beliefs of others[28,29].

We also found some evidence of a tendency for participants to believe that their peers were more optimistic than they actually are (Supplementary Fig 3). This bias was present in responses to the question about CDR and net zero, but absent for questions about peak warming and likelihood of reaching 3 °C. This could be seen as weak evidence for the hypothesis that climate scientists are biased against sharing alarming predictions[17], causing participants to perceive their peers as being slightly more optimistic than is warranted.

## Inconsistency between temperature and year of net zero $CO_2$ estimates

The median estimate for maximum warming by or before 2100 was 2.7 °C in our sample, and yet the median estimate for the year of net zero $CO_2$ (2075) is consistent with 2 °C of warming by the end of this century. This inconsistency in IPCC author responses was puzzling, and could reflect a range of underlying causes, from low confidence in scenarios to low understanding of current model estimates of the climate response to future emissions scenarios.

While our survey does not provide information with which to identify the reasoning used by the authors in answering these questions, we speculate here about five possible explanations for the differences between temperature and net zero year estimates. First, respondents may believe that climate models underestimate warming in response to emissions, for instance by underestimating climate sensitivity or the contribution of poorly understood positive climate and carbon cycle feedbacks. Second, respondents may believe that the scenario literature is not representative of the most likely emissions pathways. For example, respondents may believe that cumulative emissions of $CO_2$ leading up to when net zero $CO_2$ is reached will exceed the range that exists in the scenario literature, leading to more warming associated with a particular net zero year. Or they might believe that net zero $CO_2$ emissions will be achieved relatively early, but that warming could be higher on account of weak mitigation of non-$CO_2$ greenhouse gases or a very large warming response to decreased emissions of aerosols that currently have a cooling influence. Third, given the range of expertise present in our population (which spans expertise from paleoclimate models to urban climate adaptation), some respondents may be unfamiliar with net zero $CO_2$ dates in the scenario literature or may misunderstand the relationship between net zero emissions and resultant climate warming. Fourth, respondents may have responded more optimistically to questions with positive framing (e.g., year that net zero is achieved) compared to negative framing (e.g., maximum warming by 2100).

We also explored a fifth potential explanation involving survey wording: a small number of participants described the phrasing in our question about maximum warming as ambiguous, interpreting maximum warming before or by 2100 to potentially mean maximum conceivable warming by end of century as opposed to our intended meaning of the most likely peak or maximum temperature (see Online Methods). We investigated this possibility by plotting estimates of maximum warming against perceived likelihood of reaching 3 °C (Supplementary Fig. 1a). If a substantial number of participants were relying on this interpretation, we would expect a concentration of respondents with high estimates of peak warming but lower estimates of the likelihood of exceeding 3 °C, which was not the case. Rather, the answers to these two questions were generally consistent across the sample (e.g., most participants who estimated maximum temperature of 3 °C this century also estimating close to 50% chance of exceeding 3 °C), suggesting that most participants correctly interpreted our first question as asking for a best estimate of maximum temperature increase.

## Conclusions

IPCC Assessment Reports play a key role in shaping societal understanding of potential future climate changes. Scenarios-based model projections of future warming are a central focus of the IPCC assessment process, and the resulting ranges of future climate change are used to inform mitigation efforts as well as planning for impacts and adaptation. However, in the absence of any likelihood assessment of future scenarios, policymakers are left to use their own judgment to inform planning efforts.

Here we assess the opinions and beliefs of IPCC authors themselves, who have both societal influence and the specialist climate knowledge needed to advise decision makers. We found that most IPCC authors believe that the Paris targets are unlikely to be met and that the planet is instead on track for higher levels of warming. This finding was consistent across areas of expertise. We also found evidence of a strong relationship between IPCC authors' beliefs about future climate outcomes, and their beliefs about what

their peers believe. Thus, authors who foresee relatively higher or lower future global temperatures incorrectly believe that their own views mirror the average of the community. While this is a common and understandable way of thinking, we hope that our study presents an opportunity for members of the climate community to come to a better understanding of the beliefs of their colleagues.

Understanding the beliefs of IPCC authors can give us important information about how climate science knowledge is communicated to societal actors. The estimates of IPCC authors are not scientific certainties and we do not intend for them to be treated as simple forecasts. Rather, our participant responses provide new insights into how the IPCC community thinks and help us to better understand the beliefs that shape both the production of climate science knowledge, and the communication of this knowledge to decision makers. We hope that this study will give climate researchers a new understanding of the range of beliefs held by the greater community and catalyze open discussion amongst climate scientists about where our climate is headed.

## Methods

The authors adhered to principles of ethical research on human subjects with methods approved by the University Human Research Ethics Committee at Concordia University, Certification Number: 30016921. IPCC authors from the AR6 cycle (from the SR1.5 report onward) were invited to participate. We found email addresses for 924 living authors. Four were coauthors or early collaborators on this project, 11 had autoreply messages indicating they would not receive the invitation and five author emails could not be obtained leaving a total of 909 authors contacted. The survey was open from October 17, 2022, to December 15, 2022.

Participants were asked to predict four future climate outcomes: maximum global warming by the year 2100, likelihood of global temperatures exceeding 3 °C by 2100, year that net zero global $CO_2$ emissions are achieved, and rate of carbon dioxide removal in 2050 (see Supplementary Text for the full survey). In addition, participants were also asked to estimate the average response to each question (e.g., to estimate their peers' beliefs on all four future climate outcomes). Participants were notified that the five participants with the greatest average accuracy on these questions would be able to give $100 USD to a charity of their choosing (other participants were allotted $2 each for a selected charity). Although we asked for average estimates of peer beliefs, we report median responses here due to the non-normal distribution of responses.

Spearman rank correlations were used to measure the relationship between first- and second-order beliefs and Wilcoxon signed-rank tests were used to test for differences between medians of the first and second-order beliefs. To test for differences in perceptions of future climate outcomes between working groups, we asked participants to select the chapter of the AR6 report that best represents their area of central expertise, and then grouped participants by the working group of that chapter (thereby avoiding difficulties in assigning working groups to participants who had only participated in SR1.5 etc.). Kruskal-Wallis tests were then used to determine differences between working groups. As an exploratory test for differences between first-order (personal) and second-order (peer) estimates on the four future climate outcomes we used paired samples Wilcoxon tests. Comparisons between correlations were performed using the Fischer Z test for independent samples, and calculated in the *cocor* package[30].

Sample weights were calculated using iterative proportional fitting (raking) using the *anesrake* package[31]. Weights were calculated using gender, continent of citizenship and working group. Because participants may have contributed to multiple IPCC papers, we could not use reports authored as a weighting variable. Instead, we used their self-described chapter of expertise (e.g., participants who selected Chapter 3 of Working Group 2 were simply assigned to Working Group 2). We then selected the proportion of authors who contributed to Working Group 1, 2 or 3 as the target proportion, excluding other reports (e.g., Special Report on the Ocean and Cryosphere in a Changing Climate) from the denominator.

## Notes on outliers and data cleaning

We removed one response from the first order question on likelihood of 3 °C as it was over 100. For both questions on the year of net zero $CO_2$ we considered responses equal to or above 2400 as outliers. Some respondents provided values such as "9999" and commented that they would have preferred to select NA for certain questions or "never" as their estimated year of net zero $CO_2$, and so their large guesses indicated a qualitative disagreement with the question format. Such responses were treated as NA values. Five responses were removed from the first order estimate of year of net zero $CO_2$ and one for the second order. For both questions on the rate of CDR, we considered estimates to be outliers when they equaled or exceeded 50 $GtCO_2$ per year based on the assumption that participants may have mistaken units (e.g., assuming Mt instead of the indicated Gt). We removed six responses from the first order version of this question for that reason and seven from the second order version for this reason. We removed three values from the maximum temperature by 2100 question, one because the respondent wished to indicate NA and two because their comments on a question asking them to explain their choice indicated a misinterpretation of the question (viewing "maximum" warming to indicate the highest level of conceivable warming by 2100 instead of the best estimate of maximum temperature achieved by or before 2100). As a robustness check we performed the same statistical tests used to evaluate both hypotheses on the dataset with all outliers still included. Statistical tests were conducted in R Version 4.2.0.

## Reporting summary

Further information on research design is available in the Nature Portfolio Reporting Summary linked to this article.

## Data availability

All code and data, needed to replicate the results (e.g., anonymized survey data) are available at: https://osf.io/ytpjf/. Pre-registration available at: https://osf.io/jb526.

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

## Acknowledgements

The authors wish to thank Winston Chow for insightful feedback on this research. H.D.M. discloses support from the Natural Sciences and Engineering Research Council of Canada [RGPIN-2024-04553]. S.W. discloses support from the Social Sciences and Humanities Research Council of Canada [756-2021-0038].

## Author contributions

Seth Wynes contributed to study design, data collection, data analysis, and drafting the manuscript. Mitchell Dickau contributed to data analysis and editing the manuscript. Susan Ly contributed to data analysis and editing the manuscript. Edward Maibach, Joeri Rogelj, Kirsten Zickfeld, and Steven J Davis contributed to study design and editing the manuscript. H. Damon Matthews contributed to study design, editing the manuscript, and supervising the project.

## Competing interests

The authors declare no competing interests.
