## [Trasparent Peer Review File · Communications Earth & Environment]

Perceptions of carbon dioxide emission reductions and future warming among climate experts

Corresponding Author: Dr Seth Wynes

This manuscript has been previously reviewed at another journal. This document only contains reviewer comments, rebuttal and decision letters for versions considered at Communications Earth & Environment.

Version 0:

Decision Letter:

Dear Dr Wynes,

First of all, please allow me to apologise for the delay in sending a decision on your manuscript titled: "Perceptions of future climate outcomes among IPCC authors". It has now been seen by 3 reviewers, and we include their comments at the end of this message. They find your work of interest, but some important points are raised. We are interested in the possibility of publishing your study in Communications Earth & Environment, but would like to consider your responses to these concerns and assess a revised manuscript before we make a final decision on publication.

We therefore invite you to revise and resubmit your manuscript, along with a point-by-point response that takes into account the points raised. Please highlight all changes in the manuscript text file. In particular, for publication in Communications Earth & Environment we request that you:

*****Present novel and firmly supported insights into the range of perceptions of IPCC authors about climate scenarios and outcomes.**

*****Clarify the general theory and hypothesis and provide comprehensive details about your study sample.**

*****Consider expanding your discussion to address imbalances and limitations in your study sample, highlight critical implications, and compare and contrast your findings with previous research.**

Please use the following link to submit your revised manuscript, point-by-point response to the referees' comments (which should be in a separate document to any cover letter), a tracked-changes version of the manuscript (as a PDF file) and the completed checklist:

Link Redacted

We hope to receive your revised paper within six weeks; please let us know if you aren't able to submit it within this time so that we can discuss how best to proceed. If we don't hear from you, and the revision process takes significantly longer, we may close your file. In this event, we will still be happy to reconsider your paper at a later date, as long as nothing similar has been accepted for publication at Communications Earth & Environment or published elsewhere in the meantime.

Please do not hesitate to contact us if you have any questions or would like to discuss these revisions further. We look forward to seeing the revised manuscript and thank you for the opportunity to review your work.

Best regards,

Martina Grecequet, PhD
Associate Editor,
Communications Earth & Environment
@CommsEarth

EDITORIAL POLICIES AND FORMATTING

Editorial Policy: [Policy requirements](https://www.nature.com/documents/nr-editorial-policy-checklist.pdf) (Download the link to your computer as a PDF.)

Furthermore, please align your manuscript with our format requirements, which are summarized on the following checklist: [Communications Earth & Environment formatting checklist](https://www.nature.com/documents/commsj-phys-style-formatting-checklist-article.pdf)

and also in our style and formatting guide [Communications Earth & Environment formatting guide](https://www.nature.com/documents/commsj-phys-style-formatting-guide-accept.pdf).

*** DATA: Communications Earth & Environment endorses the principles of the Enabling FAIR data project (<http://www.copdess.org/enabling-fair-data-project/>). We ask authors to make the data that support their conclusions available in permanent, publically accessible data repositories. (Please contact the editor if you are unable to make your data available).

All Communications Earth & Environment manuscripts must include a section titled "Data Availability" at the end of the Methods section or main text (if no Methods). More information on this policy, is available at <http://www.nature.com/authors/policies/data/data-availability-statements-data-citations.pdf>.

If a community resource is unavailable, data can be submitted to generalist repositories such as [figshare](https://figshare.com/) or [Dryad Digital Repository](http://datadryad.org/). Please provide a unique identifier for the data (for example a DOI or a permanent URL) in the data availability statement, if possible. If the repository does not provide identifiers, we encourage authors to supply the search terms that will return the data. For data that have been obtained from publically available sources, please provide a URL and the specific data product name in the data availability statement. Data with a DOI should be further cited in the methods reference section.

REVIEWER COMMENTS:

Reviewer #1 (Remarks to the Author):

There has been a recent recognition of the need to undertake targeted research and surveys of relevant policymaking elites, including in the climate space. Often, these elites (whether academics or decision-makers) have a profound influence on the direction and structure of our collective climate policy responses. It is encouraging then to see the specific contribution of this manuscript - conducting a survey of IPCC authors to elicit their personal views on diverse scenarios. The authors find that their sample is pessimistic about short-term climate mitigation but optimistic about medium-term progress. The authors also show that IPCC authors show evidence that individuals assume that their peers hold similar beliefs as them (what the manuscript describes as second-order beliefs).

In general, I found the manuscript to offer an interesting data point in ongoing discussions on climate scenarios. I am open to see this type of work published but think a number of issues would need to be clarified before its findings can be of interest

to a generalist audience.

1) The paper needs to better differentiate how it expands/extends the previous IPCC surveys that it discusses (refs, 14, 18). These are not papers I have read, but the results of those papers are not explicitly contrasted / engaged here. The reader does not know if these results duplicate / corroborate / contradict / stand apart from previous IPCC survey work. While response rates are contrasted (and the lower rate may speak to fatigue by IPCC authors), we need to understand the sample comparatively and the conclusions comparatively.

2) The current sample is, of course, ambiguously representative of the total pool of IPCC authors. In general, the one balance table in the SI is inadequate treatment of this issue. It neither interprets nor contextualizes the imbalances, nor does the paper ever engage with which aspects of its findings may be conditioned or shaped by these imbalances. It is not reasonable to expect a balanced sample for elite surveys of this sort - and I agree the authors need to run with the sample they can generate. But this all has to be presented and contextualized properly. Further, while the survey doesn't include demographics, it should be possible to generate balance statistics on some demographic elements (age category, gender, academic rank) that might be further helpful in characterizing the sample against the invited population.

3) Sampling issues intersect particularly with the second-order belief analysis, which may be a major part of the authors' intended contributions here. We don't really know what group of peers each respondent was imagining when they made their estimates of second-order beliefs. Definitely they weren't imagining the particular imbalanced sample of peers who responded to this survey. So the manuscript needs to confront this by helping the reader be confident that we can assess the "quality" of the peer second-order belief data against the sample estimates of these peers' belief. At least characterizing the sampling bias (as above) would be a necessary first step here.

4) The two pre-registered hypotheses were interesting but surprised me somewhat, in that they didn't emerge organically from the theory and literature review. I would encourage the authors to set up better a general theory of why these particular hypotheses are central to our understanding of climate policy elites.

5) I found the structure of the paper confusing - particularly what seemed to be subsections of the "discussion" that iteratively repeated material previously reported in the results section, and then was itself repetitive in content to the conclusion that followed. I would recommend consolidating the entire discussion section into the results section, letting its consideration of topics like survey wording occur immediately after presenting relevant empirical results.

Reviewer #2 (Remarks to the Author):

I found the article to be very well written, clear and concise, and with a clear overarching structure. The supplementary materials provide a good level of detail of the materials and procedures used to collect the data.

I found the work of interest given the challenges faced in making societal decisions in the context of uncertainty about future climate and would like to thank the authors for undertaking this work. The insights presented demonstrate the range of perceptions about climate outcomes among the IPCC author community, which, given the authority of the IPCC, provides an interesting gauge of what this expert community thinks may happen into the future, based on their current knowledge.

The study methodology and statistical analysis looks sound, and the main claims grounded in the evidence presented, and I am therefore supportive of publication of the work.

Please note that I have been unable to access the preregistration via the link provided, as clicking the link returns an OSF webpage stating, 'You need permission'. I have therefore not been able to review the presented analyses against the pre-registration.

Major claims:

My understanding of the main claims of the article are that most IPCC authors surveyed believe (when asked to give a specific point estimate) that global warming will exceed international targets (1.5, 2.0°C) by the end of the century; that authors' first order beliefs about the likelihood of climate outcomes are positively correlated with their second order beliefs – suggesting that authors tend to consider the consensus in the author community of the climate outcomes are similar to their own beliefs; and that there are minimal differences in climate outcome estimates between authors who specialise on mitigation (WG3) and authors who specialise in impacts and adaptation (WG2).

Novelty:

The claims are, to my knowledge, novel – I'm not aware of similar work that has sought to systematically understand the IPCC author community's beliefs about future climate outcomes.

Interest to others:

I think the work will be of broad interest to the climate science and climate policy communities, and those more directly interested in studying the work of the IPCC. Future climate states will be in part determined by political and societal action (or inaction), and while it is possible to create and model scenarios of these, there is a cascade of uncertainty about what the future may hold. However, people - including policymakers and climate scientists – clearly do form judgements about what scenarios and resulting climate outcomes may or may not come to be. I think an appreciation of the IPCC's author community's views about perceptions of climate outcomes is of interest, both as a gauge from climate science experts about

what future states are considered likely in the context of international targets (which could be relevant to decision-making and policy-making), but also in terms of factors that might influence IPCC authors' views, given their role in communicating the science with society.

Evidence in support of the claims:

In my view, the major claims made in relation to the central hypotheses are well supported by the evidence presented.

Specific comments regarding other claims:

Line 198-200: From the analyses presented, it was not clear to me how the inferences about a lack of evidence about biased second order perceptions being influenced, or not, by Working Group are being made. My understanding is that the Working Group comparisons reported in the article relate to the first order beliefs (i.e. H2). Was a comparison of second order beliefs by working group conducted? or statistical comparison of the correlation coefficients between Working Groups? - i.e. do associations between first order and second order beliefs vary by working group? As you have data that could help answer this question, to support the claim made on line 198-200 I think some additional analysis here is warranted; though if doing so, suggest placing details of such analyses in supplementary materials.

Other notes about evidence presented:

Line 125-128: I found this text and Supplementary Figure 1 a bit confusing, as it was initially unclear to me why the comparisons with the IPCC scenarios were being made given the context of the research questions and hypotheses outlined in the introduction. I think some additional signposting in the writing would be helpful. I think the evidence here is really interesting and relevant, but to me it felt a bit lost in the narrative of the report up to this point, as it only really became clear to me after reading the discussion section.

Discussion / claims in context of previous literature:

Line 207-209: it was unclear to me what has informed the hypothesis stated here, i.e. that climate scientists might be biased against sharing alarming predictions. Could relevant references be included to support this possibility?

Lines 243-260: I feel the conclusion could much more directly highlight the key findings/claims of the work and the implications of these. I feel that these aspects are currently missing from this part of the text and would be helpful to readers.

Other specific but more minor comments:

Line 69: suggest clarifying what you mean by 'negative, dramatic findings', e.g. do you mean climate scientists might publicly downplay scenarios that could lead to high impact events? Similarly, may be useful to clarify what you mean by 'negative outcomes' on line 70.

Lines 92-96: it did not seem to be stated in introduction text why the hypotheses are in the directions stated. It would be useful to signpost in a sentence of two the rationale, and key concepts that informed these – for example, could signpost the false consensus effect here?

Line 105 – could a reference be provided to support the statement '... [estimates] are generally higher than estimates of the anticipated temperature outcome associated with countries meeting their stated emissions targets' ? (or is this also reference 19 as used earlier in the sentence?)

Line 288 – for transparency, for each question it would be useful to state what percentage of answers were removed for each of the data cleaning/outlier reasons stated.

Figures: for the multi-plot figures it would be useful to provide some additional space around the individual plots so that it is easier to associate the x-axis labels with the intended plot. In addition, it would be beneficial to more clearly indicate the plot labels (A, B, C etc) as the location of these varies across figures in the manuscript.

Figures: suggest slightly more descriptive axis labels to aid ease of understanding for readers, for example 'Maximum warming by 2100 (oC)'. Also some labels and captions use slightly different terminology to the questions posed to participants (e.g. figure uses 'Probability of 3oC', but question asked to ptps refers to 'Likelihood estimates of 3oC'; information sheet and question refers to 'Estimates', Figure 1 caption refers to 'Predictions'. While a minor point, it would be good to use consistent terminology to aid clarity.

Figure 1C – suggest including the years associated with the 1.5oC and 2.0oC labels for the two vertical lines.

Figure 3B - I could not see where panel B in this figure is mentioned in the results section. I think this information could be moved to supplementary materials. In addition, given the small sample sizes for some continents, it may be beneficial to provide confidence intervals or interquartile ranges for the values reported by continent. For readability and to aid comparisons, these data may be better presented in a table.

Future work: As the article indicates the survey may be repeated at future timepoints, I think it would be beneficial to ask participants about their confidence in their estimates – this would provide additional valuable information and important context to authors' estimates about future climate outcomes.

Reviewer #3 (Remarks to the Author):

The authors have conducted a survey of 211 IPCC authors from around the world, looking at their perception of the likelihood of reaching various climate outcomes. The findings are that (like people in general) scientists tend to believe their peers have similar views to themselves, and that scientists are more optimistic on some metrics (reaching net zero, carbon dioxide removal) than others (best guess for maximum warming by 2100, and likelihood of exceeding 3°C). Scientists also tend to think that their peers are more optimistic than themselves, but only with regard to reaching net zero and carbon dioxide removal.

I agree that it's interesting that IPCC authors are more pessimistic when you ask about the maximum warming before or by 2100, than they are when you ask about net zero – I think the discussion already has some good explanations for this but I also wonder if it could be related to a positive vs. negative frame? For instance, estimating maximum warming by 2100 and likelihood of exceeding 3 degrees by 2100 frame the focus on negative outcomes, whereas achieving net zero and removing CO₂ are framed in terms of more “positive” outcomes/solutions. This might also relate to why there seems to be a trend towards more optimism about peer beliefs about positive/solutions framed outcomes, but not the negatively framed outcomes.

Is there more to be said about which groups of authors are more likely to have more extreme views? E.g., by number of reports/chapters authored? By gender?

As someone who works in the area of climate and environment, this is a thought-provoking paper for me, although it is also a bleak... but given this is a communications journal and is likely to reach a wider audience, I'm curious about what the authors hope will be the impact of this paper beyond the community of climate scientists and researchers? E.g., on the discourse around climate action, on climate communication and policy.

Up to you whether you do this, but it might also be useful to make reference to other papers that have studied the role of climate scientists in climate communication, such as:

<https://doi.org/10.1007/s10584-021-03230-w>

<https://doi.org/10.1007/s10584-019-02537-z>

<http://dx.doi.org/10.1016/j.gloenvcha.2018.03.002>

https://centaur.reading.ac.uk/96012/1/23865056_Messling_Thesis.pdf

Smaller things:

Line 42-47 – It would be good to clarify what the authors mean exactly when they write about assigning probabilities to emissions scenarios – i.e., the likelihood that they will be achieved

Line 103-104 – “These responses are consistent with estimates of the climate response to current national climate policies (2.7°C by the year 2100)” – The phrasing of this is a bit confusing, it suggests you're talking about one specific country (which doesn't seem to be the case according to Fig 1. Essentially this shows that scientist predictions about warming align more with policies rather than pledges – is that right?

Communications Earth & Environment is committed to improving transparency in authorship. As part of our efforts in this direction, we are now requesting that all authors identified as ‘corresponding author’ create and link their Open Researcher and Contributor Identifier (ORCID) with their account on the Manuscript Tracking System prior to acceptance. ORCID helps the scientific community achieve unambiguous attribution of all scholarly contributions. You can create and link your ORCID from the home page of the Manuscript Tracking System by clicking on ‘Modify my Springer Nature account’ and following the instructions in the link below. Please also inform all co-authors that they can add their ORCIDs to their accounts and that they must do so prior to acceptance.

Author Rebuttal letter:

The author's response to these comments can be found at the end of this file.

Version 1:

Decision Letter:

Dear Dr Wynes,

Your manuscript titled "Perceptions of future climate outcomes among IPCC authors" has now been seen by our reviewers, whose comments appear below. In light of their advice we are delighted to say that we are happy, in principle, to publish a suitably revised version in Communications Earth & Environment under the open access CC BY license (Creative Commons Attribution v4.0 International License).

We therefore invite you to revise your paper one last time to address the remaining concerns of our reviewers. At the same time we ask that you edit your manuscript to comply with our format requirements and to maximise the accessibility and therefore the impact of your work.

EDITORIAL REQUESTS:

****Please take care to match our formatting and policy requirements. We will check revised manuscript and return manuscripts that do not comply. Such requests will lead to delays. ****

SUBMISSION INFORMATION:

OPEN ACCESS:

Communications Earth & Environment is a fully open access journal. Articles are made freely accessible on publication under a [CC BY license](http://creativecommons.org/licenses/by/4.0) (Creative Commons Attribution 4.0 International License). This license allows maximum dissemination and re-use of open access materials and is preferred by many research funding bodies.

For further information about article processing charges, open access funding, and advice and support from Nature Research, please visit <https://www.nature.com/commsenv/article-processing-charges>

At acceptance, you will be provided with instructions for completing this CC BY license on behalf of all authors. This grants us the necessary permissions to publish your paper. Additionally, you will be asked to declare that all required third party permissions have been obtained, and to provide billing information in order to pay the article-processing charge (APC).

Link Redacted

Best regards,

Martina Grecequet, PhD
Associate Editor,
Communications Earth & Environment

@CommsEarth

REVIEWERS' COMMENTS:

Reviewer #1 (Remarks to the Author):

The authors have done a good job addressing the comments I raised in my initial review. The new empirical details and theoretical context all help the reader understand the significance of the findings and the generalizability of the findings.

While I still would prefer a more integrated results/discussion format, I appreciate the author's openness to this and defer to the editor and other reviewers who were happy with the existing format.

Reviewer #2 (Remarks to the Author):

Thank you for your careful consideration of the review comments and questions. I have worked through the revised documents, and am satisfied that my comments have been appropriately considered and addressed.

With regards to accessing the pre-registration, the web link provided in the manuscript took me to the project file repository, but I could not locate the pre-registration details there.

However, I was able to locate the pre-reg via OSF search, located here: <https://doi.org/10.17605/OSF.IO/JB526> - perhaps the url on line 98 needs updating to this one?

Reviewer #3 (Remarks to the Author):

I'm largely satisfied with the changes that the authors have made to the manuscript and thank them for addressing my feedback. There are two additional points that I'd like to ask the authors to consider/address:

1) In the time since the last review, this Guardian article/survey has been published showing similar findings. It might be worth making reference to it in some way. Link: <https://www.theguardian.com/environment/article/2024/may/08/world-scientists-climate-failure-survey-global-temperature>

2) And while on the whole I agree with the framing of the paper, there are certain lines where I think that the role of IPCC scientist predictions may be overstated, e.g., "However, in the absence of any likelihood assessment of future scenarios, policymakers are left to use their own non-expert judgement to inform planning efforts". Here, and also in other areas linking to broader context (introduction paragraph), it is worth clarifying that the opinions of IPCC scientists about likely warming/other outcomes are not scientific fact, or necessarily more objective than the opinions of policymakers or other stakeholders.

This might seem obvious, but while the survey questions relate to IPCC scientists' expertise, these kinds of judgments also encompass elements that fall outside of climate science, including views about political contexts and decision-making, geopolitics, public opinion, policy implementation, and so on - and here scientist estimations can't necessarily be seen to be more 'expert' than policymakers' own judgments, or anyone else's. And the influence that IPCC scientists have means it's even more important to be careful about how these judgements of likelihood are interpreted and communicated.

Author Rebuttal letter:

Response to Reviewer Comments

REVIEWERS' COMMENTS:

Reviewer #1 (Remarks to the Author):

The authors have done a good job addressing the comments I raised in my initial review. The new empirical details and theoretical context all help the reader understand the significance of the findings and the generalizability of the findings.

While I still would prefer a more integrated results/discussion format, I appreciate the author's openness to this and defer to the editor and other reviewers who were happy with the existing

format.

Thank you again for your insightful additions to this manuscript.

Reviewer #2 (Remarks to the Author):

Thank you for your careful consideration of the review comments and questions. I have worked through the revised documents, and am satisfied that my comments have been appropriately considered and addressed.

Thank you for taking the time to comment again.

With regards to accessing the pre-registration, the web link provided in the manuscript took me to the project file repository, but I could not locate the pre-registration details there. However, I was able to locate the pre-reg via OSF search, located here: <https://doi.org/10.17605/OSF.IO/JB526> - perhaps the url on line 98 needs updating to this one?

You are correct. This URL has been added to the paper on line 98.

Reviewer #3 (Remarks to the Author):

I'm largely satisfied with the changes that the authors have made to the manuscript and thank them for addressing my feedback.

Thank you again for your suggestions.

There are two additional points that I'd like to ask the authors to consider/address:

1) In the time since the last review, this Guardian article/survey has been published showing similar findings. It might be worth making reference to it in some way.

Link: <https://www.theguardian.com/environment/article/2024/may/08/world-scientists-climate-failure-survey-global-temperature>

A reference to this article is now made on Line 206 which now reads, "A previous survey found 60% of Working Group 1 authors believed that warming of 3°C or more is likely by 2100, while another found that 77% of IPCC authors and editors expected at least 2.5°C in the same timeframe."

2) And while on the whole I agree with the framing of the paper, there are certain lines where I think that the role of IPCC scientist predictions may be overstated, e.g., "However, in the absence of any likelihood assessment of future scenarios, policymakers are left to use their own non-expert judgement to inform planning efforts". Here, and also in other areas linking to broader context (introduction paragraph), it is worth clarifying that the opinions of IPCC scientists about likely warming/other outcomes are not scientific fact, or necessarily more objective than the opinions of policymakers or other stakeholders.

This might seem obvious, but while the survey questions relate to IPCC scientists' expertise, these kinds of judgments also encompass elements that fall outside of climate science, including views about political contexts and decision-making, geopolitics, public opinion, policy implementation, and so on - and here scientist estimations can't necessarily be seen to be more 'expert' than policymakers' own judgments, or anyone else's. And the influence that IPCC scientists have means it's even more important to be careful about how these judgements of likelihood are interpreted and communicated.

We have made several changes in an effort to use more cautious wording. For

instance, line 45 now reads "Furthermore, the absence of likelihood information about future climate outcomes means that the responsibility to select the most relevant scenarios for impacts and adaptation planning is transferred from climate scientists to policymakers^{6,7}."

Previously the text read "Furthermore, the absence of likelihood information about future climate outcomes means that the responsibility to select the most relevant scenarios for impacts and adaptation planning is transferred from highly qualified climate scientists to less qualified policymakers^{6,7}."

We have also revised the text in question to read: "However, in the absence of any likelihood assessment of future scenarios, policymakers are left to use their own judgement to inform planning efforts."

REVIEWER COMMENTS:

Reviewer #1 (Remarks to the Author):

There has been a recent recognition of the need to undertake targeted research and surveys of relevant policymaking elites, including in the climate space. Often, these elites (whether academics or decision-makers) have a profound influence on the direction and structure of our collective climate policy responses. It is encouraging then to see the specific contribution of this manuscript - conducting a survey of IPCC authors to elicit their personal views on diverse scenarios. The authors find that their sample is pessimistic about short-term climate mitigation but optimistic about medium-term progress. The authors also show that IPCC authors show evidence that individuals assume that their peers hold similar beliefs as them (what the manuscript describes as second-order beliefs).

In general, I found the manuscript to offer an interesting data point in ongoing discussions on climate scenarios. I am open to see this type of work published but think a number of issues would need to be clarified before its findings can be of interest to a generalist audience.

Thank you for taking the time to read our paper and offer valuable insights.

1) The paper needs to better differentiate how it expands/extends the previous IPCC surveys that it discusses (refs, 14, 18). These are not papers I have read, but the results of those papers are not explicitly contrasted / engaged here. The reader does not know if these results duplicate / corroborate / contradict / stand apart from previous IPCC survey work. While response rates are contrasted (and the lower rate may speak to fatigue by IPCC authors), we need to understand the sample comparatively and the conclusions comparatively.

The other peer reviewed research on this topic does not overlap substantially with our work. The non-peer reviewed survey by Nature includes a single question about future climate outcomes, which broadly agrees with our findings. We have added the following section to the Discussion to give readers this context (Line 221):

A previous survey found 60% of Working Group 1 authors believed that warming of 3°C or more is likely by 2100¹⁸. Our results corroborate that there is a widespread belief among IPCC authors that substantial warming is likely before 2100, but also extends these findings by including authors from all working groups, asking about additional climate outcomes, and shedding light on second order beliefs. Other surveys of IPCC authors have also been conducted, focusing for instance on gender bias in the IPCC¹⁴ and science communication¹².

2) The current sample is, of course, ambiguously representative of the total pool of IPCC authors. In general, the one balance table in the SI is inadequate treatment of this issue. It neither interprets nor contextualizes the imbalances, nor does the paper ever engage with which aspects of its findings may be conditioned or shaped by these imbalances. It is not reasonable

to expect a balanced sample for elite surveys of this sort - and I agree the authors need to run with the sample they can generate. But this all has to be presented and contextualized properly. Further, while the survey doesn't include demographics, it should be possible to generate balance statistics on some demographic elements (age category, gender, academic rank) that might be further helpful in characterizing the sample against the invited population.

Thank you for this feedback. We conducted additional analysis, described our results in the main section of the paper, and reported the detailed statistics in the Supplement. Line 130 now reads:

“Although we have presented unweighted measures here, we also checked to see if issues with the representativeness of our survey affected the results. We found that when our sample is weighted to reflect the demographics of the IPCC population as a whole (using proportional iterative weighting, or raking, according to gender, the continent of citizenship, and IPCC Working Group), the mean responses do not fluctuate substantially. For instance, the unweighted mean of estimated year of net zero CO₂ is 2090, whereas the weighted mean is 2088 (see Supplementary Table 2 for full results).”

We also raise this as a potential limitation in the Discussion (Line 238):

“One potential critique of this finding is that while we asked participants to estimate the average response of their peers, participants may have had in mind the actual population of IPCC researchers, and yet our sample is not perfectly representative of the population. We tried to address this by weighting the sample and found that weighted means of future climate estimates were remarkably similar to unweighted means (Supplementary Table 2). Realistically however, we would acknowledge that the more likely cognitive process for a participant trying to estimate the beliefs of the entire population would be to rely on the beliefs of near-peers, whose ideas would be more cognitively available than objective indicators (which are rare outside of this study).”

Unfortunately, we do not have access to a broad range of demographic information, including questions about participant age or academic rank. We intentionally designed our survey to be quite brief to remove as many impediments as possible for a group of participants who are already overburdened. We did ask questions about gender, continent, and which reports participants have participated in. Statistics showing the proportions of the sample and the population on these demographics are available in the Supplement.

3) Sampling issues intersect particularly with the second-order belief analysis, which may be a major part of the authors' intended contributions here. We don't really know what group of peers each respondent was imagining when they made their estimates of second-order beliefs. Definitely they weren't imagining the particular imbalanced sample of peers who responded to this survey. So the manuscript needs to confront this by helping the reader be confident that we can assess the "quality" of the peer second-order belief data against the sample estimates of

these peers' belief. At least characterizing the sampling bias (as above) would be a necessary first step here.

The additional results discussed above and added to the Supplement show that differences in estimates between a sample weighted according to the population of IPCC researchers are negligibly different from those that we originally reported, giving us confidence that even if participants were estimating the population as a whole, instead of the unbalanced sample that we were able to gather, the difference is not meaningful.

4) The two pre-registered hypotheses were interesting but surprised me somewhat, in that they didn't emerge organically from the theory and literature review. I would encourage the authors to set up better a general theory of why these particular hypotheses are central to our understanding of climate policy elites.

We have added to the text (Line 97):

“Before distributing the survey, we preregistered two hypotheses (https://osf.io/ytpjf/?view_only=35744a77f4584665a8acc142f2545905):

H1) There will be a positive correlation between estimates of peer beliefs and personal estimates of future climate outcomes.

This hypothesis was based on research in other domains demonstrating a “false consensus effect”, where people believe their own judgments to be common and opposing judgments to be uncommon^{19,20}.

H2) Researchers with a focus on climate solutions (Working Group 3) will have more optimistic perceptions of future climate outcomes than those who work on climate impacts and adaptation (Working Group 2).

We expected that researchers working on climate solutions would have greater familiarity with recent literature suggesting that the worst-case climate outcomes have become increasingly unlikely^{21,22} and greater familiarity with the rapidly improving solution set. Furthermore, because participants were generating rapid responses rather than carefully crafted analysis, we would expect them to rely on the availability heuristic (where people judge the probability of events by how readily they come to mind²³) which would make researchers who work on climate solutions more prone to accessing optimistic estimates.”

5) I found the structure of the paper confusing - particularly what seemed to be subsections of the "discussion" that iteratively repeated material previously reported in the results section, and then was itself repetitive in content to the conclusion that followed. I would recommend consolidating the entire discussion section into the results section, letting its consideration of topics like survey wording occur immediately after presenting relevant empirical results.

Thank you for sharing this suggestion. After some attempts at reorganization, we felt that the paper was better suited to having distinct Results and Discussion sections despite the fact that this necessarily introduces some repetition.

Reviewer #2 (Remarks to the Author):

I found the article to be very well written, clear and concise, and with a clear overarching structure. The supplementary materials provide a good level of detail of the materials and procedures used to collect the data.

Thank you!

I found the work of interest given the challenges faced in making societal decisions in the context of uncertainty about future climate and would like to thank the authors for undertaking this work. The insights presented demonstrate the range of perceptions about climate outcomes among the IPCC author community, which, given the authority of the IPCC, provides an interesting gauge of what this expert community thinks may happen into the future, based on their current knowledge.

The study methodology and statistical analysis looks sound, and the main claims grounded in the evidence presented, and I am therefore supportive of publication of the work.

Please note that I have been unable to access the preregistration via the link provided, as clicking the link returns an OSF webpage stating, 'You need permission'. I have therefore not been able to review the presented analyses against the pre-registration.

**Our apologies for this. The link should now be available at:
https://osf.io/ytpjf/?view_only=35744a77f4584665a8acc142f2545905**

Major claims:

My understanding of the main claims of the article are that most IPCC authors surveyed believe (when asked to give a specific point estimate) that global warming will exceed international targets (1.5, 2.0OC) by the end of the century; that authors' first order beliefs about the likelihood of climate outcomes are positively correlated with their second order beliefs – suggesting that authors tend to consider the consensus in the author community of the climate outcomes are similar to their own beliefs; and that there are minimal differences in climate outcome estimates between authors who specialise on mitigation (WG3) and authors who

specialise in impacts and adaptation (WG2).

Novelty:

The claims are, to my knowledge, novel – I'm not aware of similar work that has sought to systematically understand the IPCC author community's beliefs about future climate outcomes.

Interest to others:

I think the work will be of broad interest to the climate science and climate policy communities, and those more directly interested in studying the work of the IPCC. Future climate states will be in part determined by political and societal action (or inaction), and while it is possible to create and model scenarios of these, there is a cascade of uncertainty about what the future may hold. However, people - including policymakers and climate scientists – clearly do form judgements about what scenarios and resulting climate outcomes may or may not come to be. I think an appreciation of the IPCC's author community's views about perceptions of climate outcomes is of interest, both as a gauge from climate science experts about what future states are considered likely in the context of international targets (which could be relevant to decision-making and policy-making), but also in terms of factors that might influence IPCC authors' views, given their role in communicating the science with society.

Evidence in support of the claims:

In my view, the major claims made in relation to the central hypotheses are well supported by the evidence presented.

Specific comments regarding other claims:

Line 198-200: From the analyses presented, it was not clear to me how the inferences about a lack of evidence about biased second order perceptions being influenced, or not, by Working Group are being made. My understanding is that the Working Group comparisons reported in the article relate to the first order beliefs (i.e. H2). Was a comparison of second order beliefs by working group conducted? or statistical comparison of the correlation coefficients between Working Groups? - i.e. do associations between first order and second order beliefs vary by working group? As you have data that could help answer this question, to support the claim made on line 198-200 I think some additional analysis here is warranted; though if doing so, suggest placing details of such analyses in supplementary materials.

Thank you for bringing this to our attention.

For the reviewers' benefit, a visual analysis showing the relationships between first and second order perceptions according to working group reveals no striking differences (added below). We have added a table of correlation coefficients for each working group and each pair of future outcomes to the Supplement. We have also conducted the suggested statistical analysis and added to the text (Line 198):

“We had no hypothesis regarding whether some working groups would have a higher correlation between their first and secondary beliefs than others. However, we tested working group-specific correlations and found that all working groups displayed similar correlations between primary and secondary beliefs on all four future climate outcomes (Supplementary Table 3). The largest differences were

observed between Working Group 2 and Working Group 3, but in a statistical comparison of those correlations, none were found to be significantly different. We also did not register a hypothesis regarding differences between continents, but have made a breakdown of results by continent available for the interested reader (Supplementary Table 4).”

Other notes about evidence presented:

Line 125-128: I found this text and Supplementary Figure 1 a bit confusing, as it was initially unclear to me why the comparisons with the IPCC scenarios were being made given the context of the research questions and hypotheses outlined in the introduction. I think some additional signposting in the writing would be helpful. I think the evidence here is really interesting and relevant, but to me it felt a bit lost in the narrative of the report up to this point, as it only really became clear to me after reading the discussion section.

We understand how this might have caused confusion. We have added to the Introduction (Line 83):

“Asking for estimates of four future climate outcomes had two benefits: it increased certainty in the understanding of the overall outlook of participants, and also illuminated differences in the way that IPCC authors understand potential futures. For instance, if participants anticipate high temperatures by 2100, but also anticipate net zero CO₂ being reached relatively early in the century, it could

indicate beliefs about the difficulty in mitigating non-CO₂ gases which might be explored further in a later survey.”

Discussion / claims in context of previous literature:

Line 207-209: it was unclear to me what has informed the hypothesis stated here, i.e. that climate scientists might be biased against sharing alarming predictions. Could relevant references be included to support this possibility?

Thanks for pointing this out. The relevant reference provided earlier in the text, which has now been added to this line as well, is from Brysse et al. (2013) Global Environmental Change “Climate change prediction: Erring on the side of least drama?”

Lines 243-260: I feel the conclusion could much more directly highlight the key findings/claims of the work and the implications of these. I feel that these aspects are currently missing from this part of the text and would be helpful to readers.

We agree that the Conclusion was overly sparse on details. We hope that the following addition better highlights the key claims:

“We found that most IPCC authors believe that the Paris targets are unlikely to be met and that the planet is instead on track for higher levels of warming. This finding was consistent across areas of expertise. We also found evidence of a strong relationship between IPCC authors’ beliefs about future climate outcomes, and their beliefs about what their peers believe. Thus, authors who foresee relatively higher or lower future global temperatures incorrectly believe that their own views mirror the average of the community. While this is a common and understandable way of thinking, we hope that our study presents an opportunity for members of the climate community to come to a better understanding of the beliefs of their colleagues.”

Other specific but more minor comments:

Line 69: suggest clarifying what you mean by ‘negative, dramatic findings’, e.g. do you mean climate scientists might publicly downplay scenarios that could lead to high impact events? Similarly, may be useful to clarify what you mean by ‘negative outcomes’ on line 70.

We have added detail and rephrased. The lines now read: “This could potentially be problematic if climate scientists feel pressured to abide by norms of restraint and therefore publicly downplay negative, dramatic findings¹⁷. If this then caused other members of the community to underestimate the likelihood of negative outcomes, they might in turn rely on those faulty assumptions in their own research decisions or public communications.

Lines 92-96: it did not seem to be stated in introduction text why the hypotheses are in the directions stated. It would be useful to signpost in a sentence of two the rationale, and key concepts that informed these – for example, could signpost the false consensus effect here?

Thanks for this suggestion. The text now reads (Line 97):

“Before distributing the survey, we preregistered two hypotheses (https://osf.io/ytpjf/?view_only=35744a77f4584665a8acc142f2545905):

H1) There will be a positive correlation between estimates of peer beliefs and personal estimates of future climate outcomes.

This hypothesis was based on research in other domains demonstrating a “false consensus effect”, where people believe their own judgments to be common and opposing judgments to be uncommon^{19,20}.

H2) Researchers with a focus on climate solutions (Working Group 3) will have more optimistic perceptions of future climate outcomes than those who work on climate impacts and adaptation (Working Group 2).

We expected that researchers working on climate solutions would have greater familiarity with recent literature suggesting that the worst-case climate outcomes have become increasingly unlikely^{21,22} and greater familiarity with the rapidly improving solution set. Furthermore, because participants were generating rapid responses rather than carefully crafted analysis, we would expect them to rely on the availability heuristic (where people judge the probability of events by how readily they come to mind²³) which would make researchers who work on climate solutions more prone to accessing optimistic estimates.”

Line 105 – could a reference be provided to support the statement ‘... [estimates] are generally higher than estimates of the anticipated temperature outcome associated with countries meeting their stated emissions targets’ ? (or is this also reference 19 as used earlier in the sentence?)

That’s correct – it is the same reference.

Line 288 – for transparency, for each question it would be useful to state what percentage of

answers were removed for each of the data cleaning/outlier reasons stated.

We now indicate in the text the number of responses cleaned from these questions.

Figures: for the multi-plot figures it would be useful to provide some additional space around the individual plots so that it is easier to associate the x-axis labels with the intended plot. In addition, it would be beneficial to more clearly indicate the plot labels (A, B, C etc) as the location of these varies across figures in the manuscript.

Following subsequent feedback, we deleted the world map from the plot so that the figures now follow the same consistent theme in terms of labelling. Labels were added to Figure 3 such that A indicates the plot in the top left, B indicates top right etc. which is the same throughout, hopefully aiding in reader comprehension.

Figures: suggest slightly more descriptive axis labels to aid ease of understanding for readers, for example 'Maximum warming by 2100 (oC)'. Also some labels and captions use slightly different terminology to the questions posed to participants (e.g. figure uses 'Probability of 3oC', but question asked to ptpts refers to 'Likelihood estimates of 3oC'; information sheet and question refers to 'Estimates', Figure 1 caption refers to 'Predictions'. While a minor point, it would be good to use consistent terminology to aid clarity.

We have revised the paper to use more descriptive and consistent terms for these variables including in labels.

Figure 1C – suggest including the years associated with the 1.5oC and 2.0oC labels for the two vertical lines.

Years are now indicated in the figure caption.

Figure 3B - I could not see where panel B in this figure is mentioned in the results section. I think this information could be moved to supplementary materials. In addition, given the small sample sizes for some continents, it may be beneficial to provide confidence intervals or interquartile ranges for the values reported by continent. For readability and to aid comparisons, these data may be better presented in a table.

As suggested, we have moved this data to the Supplement in the form of a table and added IQR values.

Future work: As the article indicates the survey may be repeated at future timepoints, I think it would be beneficial to ask participants about their confidence in their estimates – this would provide additional valuable information and important context to authors' estimates about future climate outcomes.

We appreciate this suggestion!

Reviewer #3 (Remarks to the Author):

The authors have conducted a survey of 211 IPCC authors from around the world, looking at their perception of the likelihood of reaching various climate outcomes. The findings are that (like people in general) scientists tend to believe their peers have similar views to themselves, and that scientists are more optimistic on some metrics (reaching net zero, carbon dioxide removal) than others (best guess for maximum warming by 2100, and likelihood of exceeding 3°C). Scientists also tend to think that their peers are more optimistic than themselves, but only with regard to reaching net zero and carbon dioxide removal.

Thank you for your close reading of our paper!

I agree that it's interesting that IPCC authors are more pessimistic when you ask about the maximum warming before or by 2100, than they are when you ask about net zero – I think the discussion already has some good explanations for this but I also wonder if it could be related to a positive vs. negative frame? For instance, estimating maximum warming by 2100 and likelihood of exceeding 3 degrees by 2100 frame the focus on negative outcomes, whereas achieving net zero and removing CO2 are framed in terms of more "positive" outcomes/solutions. This might also relate to why there seems to be a trend towards more optimism about peer beliefs about positive/solutions framed outcomes, but not the negatively framed outcomes.

This is an interesting hypothesis. We have added to the text:

“Fourth, respondents may have responded more optimistically to questions with positive framing (e.g. year that net zero is achieved) compared to negative framing (e.g. maximum warming by 2100).”

Is there more to be said about which groups of authors are more likely to have more extreme views? E.g., by number of reports/chapters authored? By gender?

We performed some additional analysis on these questions and visualizing the results leads us to believe there is no strong evidence to support the idea that one group is more likely to have extreme views than another. We have included

visualizations below for gender, continent and working group. There is some indication that women and men have different predictions for the likelihood of 3C. This was the only significant relationship according to a t-test out of the four future climate outcome predictions ($t = 2.2$, $df = 88.328$, $p\text{-value} = 0.03041$, mean in group Female = 56.65, mean in group Male = 48.31). However, since we made multiple comparisons (and without pre-registration) this would not be significant when accounting for multiple comparisons. Furthermore, we previously rejected a hypothesis for only having a significant result in one out of the four questions, and so we do not believe there is enough evidence to build a case for a conclusion here.

We created a composite variable which we will call “seniority” that sums the total number of reports each participant contributed to. Visually we saw no evidence of extreme estimates being related to seniority, but because only a very small number of participants contributed to many reports (e.g. 5+) it might be possible to identify their responses from the data if visualized. We therefore simply report correlations. None were significant.

Seniority vs Maximum temperature by 2100: $r_s=0.08$, $p=.46$

Seniority vs Predicted Probability of 3C: $r_s=-0.05$, $p=.22$

Seniority vs Year of Net Zero: $r_s=-0.14$, $p=.051$

Seniority vs Rate of CDR: $r_s=-0.05$, $p=.60$

As someone who works in the area of climate and environment, this is a thought-provoking paper for me, although it is also a bleak... but given this is a communications journal and is likely to reach a wider audience, I'm curious about what the authors hope will be the impact of this paper beyond the community of climate scientists and researchers? E.g., on the discourse around climate action, on climate communication and policy.

Primarily we hope that this paper will serve as a useful opportunity for climate experts to reassess what they believe in light of having a clearer understanding of the beliefs of their peers. We would also hope that it would serve as a useful datapoint for decision makers and everyday members of the public seeking to better understand what future climate outcomes are most likely.

Up to you whether you do this, but it might also be useful to make reference to other papers that have studied the role of climate scientists in climate communication, such as:

<https://doi.org/10.1007/s10584-021-03230-w>

<https://doi.org/10.1007/s10584-019-02537-z>

<http://dx.doi.org/10.1016/j.gloenvcha.2018.03.002>

https://centaur.reading.ac.uk/96012/1/23865056_Messling_Thesis.pdf

Thank you for sharing. We have added content to the Introduction based on the research from the first paper. The revised section (Line 59) now reads:

“Furthermore, many IPCC authors engage both directly and indirectly (through media interviews) in efforts to share their knowledge with the public and policymakers¹², which sometimes includes sharing their professional beliefs on a range of speculative topics, including the feasibility of achieving temperature targets¹³.

Smaller things:

Line 42-47 – It would be good to clarify what the authors mean exactly when they write about assigning probabilities to emissions scenarios – i.e., the likelihood that they will be achieved

We have modified the text so that it now reads, “The IPCC chooses not to describe the probability of different scenarios occurring,”

Line 103-104 – “These responses are consistent with estimates of the climate response to current national climate policies (2.7°C by the year 2100)” – The phrasing of this is a bit confusing, it suggests you’re talking about one specific country (which doesn’t seem to be the case according to Fig 1. Essentially this shows that scientist predictions about warming align more with policies rather than pledges – is that right?

You are correct – scientist predictions align closer with the outcomes associated with models of current policies rather than pledges.

We have rephrased to avoid confusion. The text now reads (Line 120): “These participant estimates are consistent with modeling outputs of the climate response to current national climate policies (2.7°C by the year 2100²⁴), and are generally higher than estimates of the anticipated temperature outcome associated with countries meeting their stated emissions targets.